# Innovative HDPE Composites Enriched with UV Stabilizer and Diatomaceous Earth/Zinc Oxide for Enhanced Seafood Packaging and Antimicrobial Properties

**DOI:** 10.3390/polym15234577

**Published:** 2023-11-30

**Authors:** Korakot Charoensri, Yang J. Shin, Hyun J. Park

**Affiliations:** Department of Biotechnology, College of Life Sciences and Biotechnology, Korea University, 145 Anam-ro, Seongbuk-gu, Seoul 02841, Republic of Korea; korapop253@korea.ac.kr

**Keywords:** active packaging, food packaging, seafood, antimicrobial, zinc oxide, UV stabilizers

## Abstract

The fisheries industry encounters distinct packaging challenges, including the need to protect perishable seafood from rapid spoilage caused by UV radiation while allowing for reuse. This study tackles these issues by introducing advanced high-density polyethylene (HDPE) composites enhanced with a UV stabilizer and inorganic fillers, such as diatomaceous earth/zinc oxide (DZ). Our investigation explores the transformative effects of weathering on these pioneering composites, evaluating shifts in mechanical, physical, thermal properties, and sub-zero temperature stability. Incorporating a UV stabilizer alongside DZ within the HDPE matrix significantly enhances mechanical performance and weathering resilience. These enhancements extend the longevity of seafood packaging while preserving product quality. Moreover, our findings reveal a substantial breakthrough in antimicrobial properties. The inclusion of DZ, with or without a UV stabilizer, results in an impressive up to 99% enhancement in antibacterial activity against both Gram-positive and Gram-negative bacteria. This discovery not only bolsters the protective attributes of HDPE packaging but also presents a compelling case for the development of active packaging materials derived from DE/ZnO composites. This study bridges the gap between packaging and seafood quality, introducing advanced polymeric packaging technology for seafood products. It highlights the mutually beneficial link between packaging improvements and ensuring seafood quality, meeting industry needs while promoting sustainability.

## 1. Introduction

The packaging of seafood products has emerged as a pivotal factor within the global seafood industry, signifying the paramount need to uphold product quality. Characteristics such as color, odor, texture, and nutritional content significantly influence the quality and consumer acceptance of seafood products [1,2,3]. In light of the potential drawbacks affecting seafood items, the imperative arises to devise innovative technologies aimed at enhancing and regulating their quality. Central to this pursuit is the continuous advancement in packaging science and technology. A critical challenge facing seafood products pertains to lipid photooxidation, particularly evident in fish, prawns, and shrimps, leading to undesirable outcomes such as off-flavor, rancidity, color deterioration, aldehyde formation, and microbial proliferation [1,4,5]. To ensure sustained growth in the seafood product market, it becomes essential to introduce and develop active packaging solutions with high-performance antimicrobial materials capable of reducing off-flavor and blocking harmful UV rays [6,7,8,9]. Zinc oxide (ZnO) has emerged as a promising candidate for active packaging development owing to its dual functionality as a UV-blocking and antimicrobial agent [10,11,12,13,14,15]. Numerous reports substantiate ZnO’s antimicrobial efficacy against both Gram-positive and Gram-negative bacteria. Proposed mechanisms for ZnO’s antibacterial properties encompass metal ion release, interactions between ZnO nanoparticles and microorganisms, and the generation of reactive oxygen species (ROS) under light radiation [13,16,17,18]. ZnO’s exceptional UV-blocking capabilities stem from its wide band gap (Eg = 3.37 eV, corresponding to 376 nm), offering unique electro-optical properties and effective UV absorption while maintaining visible light transparency [15,19,20,21,22]. Importantly, ZnO is renowned for its stability within polymeric matrices, displaying photostability and thermal resilience, thereby conferring longevity and non-migratory attributes [21,23,24,25]. This study endeavors to yield an active packaging material with the potential for real-world application in the Korean seafood product market, characterized by outstanding antibacterial performance, off-flavor mitigation, stability, and reusability. To enhance ZnO’s attributes, we have incorporated diatomaceous earth (DE) based on our previous research [26], with the aim of augmenting antimicrobial efficacy through increased surface area to volume ratio, consequently bolstering production due to its microporous structure [27,28,29]. Furthermore, we have introduced Tinnuvin 770 (T770) (Bis 12, 2, 6, 6-tetramethyl-4-piperidyl) sebecate as a solid basic amine light stabilizer (HALS) to enhance ZnO’s UV stability [30,31]. This addition has been investigated to ascertain the improved properties of the active packaging material specimens. In this comprehensive study, we evaluate injection-molded HDPE/DE/ZnO ternary compositions with varying relative UV stabilizer proportions, focusing on their antibacterial activity, capacity to mitigate off-flavor, stability under sub-zero temperatures, and weathering resilience. These findings are poised to contribute to the advancement of the seafood product packaging industry, aligning it with the escalating demand for seafood products within the food industry.

## 2. Materials and Methods

### 2.1. Materials

Zinc chloride (ZnCl_2_) was procured from Sam Chun Pure Chemicals in Gyeonggi, Republic of Korea. The commercial DE (Celite—CF-1031) was obtained from Tianjin Chemist Scientific, Ltd. in Tianjin, China. Sodium hydroxide (NaOH) was acquired from Daejung Chemicals Metals, also located in Gyeonggi, Republic of Korea. The commercial HDPE resin used in the study exhibited a melt flow rate of 4 g/10 min at 190 °C and 2.16 kg, along with a density of 0.955 g/cm^3^. Additionally, bis(2,2,6,6-tetramethyl-4-piperidyl)sebacate, commonly known as Tinuvin 770, was sourced from Aldrich Chemical Co. (Shanghai, China).

### 2.2. Preparation of Composite and Injection Molding

In accordance with our previously reported findings [26], the composite consisting of 4% wt diatomaceous earth and zinc oxide (DE/ZnO, hereafter referred to as DZ) in high-density polyethylene (HDPE) exhibited superior properties for use as an active packaging material for seafood products. Consequently, this composite was selected as the base material for the current study. To enhance its performance and UV resistance, T770 was introduced as a UV stabilizer, and the blending process was conducted using a planetary mixer. The UV stabilizer was carefully mixed with the prepared masterbatch at specified amounts of 500, 1000, and 2000 ppm, resulting in the formulation of three distinct samples, denoted as HDPE/DZ/T-500, HDPE/DZ/T-1000, and HDPE/DZ/T-2000. The extrusion process was carried out using a twin-screw extruder (LTE-20-40, Labtech Engineering, Samut Prakan, Thailand) operating within the temperature range of 80–150 °C and a rotation speed of 180 rpm. Subsequently, the extrudates were precision-cut into 2.5 mm segments utilizing a palletizer (LZ-120, Labtech Engineering). These resulting pellets were subsequently molded into test specimens using an injection machine (Meteor 270/75, Mateo & Sole, Barcelona, Spain) equipped with a mirror-finishing steel mold featuring standard geometries for sample characterization. The injection process applied a dampening force of 75 tons, with cavity filling and cooling durations set at 1 and 10 s, respectively.

### 2.3. Characterization

The surface morphology and elemental composition of the active packaging material specimens were analyzed through field-emission scanning electron microscopy (FE-SEM) employing the Hitachi 508010 instrument from Tokyo, Japan. Additionally, energy dispersive spectroscopy (EDS) mapping was utilized to elucidate the elemental distribution. To assess the thermal stability of the test specimens, thermogravimetric analysis (TGA) was performed using the SCINCO N-100 instrument in Seoul, Republic of Korea. Specimens, with weights falling within the range of 15–20 mg, underwent scanning from 100 °C to 600 °C at a heating rate of 10 °C/min under a nitrogen atmosphere with a flow rate of 30 mL/min. The maximum degradation rate temperature was determined by identifying the peak in the first derivative (DTG) of the TGA curve. In order to evaluate the mechanical properties of the material and ascertain its suitability for seafood product packaging, Izod impact tests and brittleness temperature assessments were conducted. Izod tests were executed utilizing Instron LEAST 9050 (Norwood, MA, USA) impact pendulums equipped with a 0.5 J hammer, adhering to the guidelines outlined in ASTM D746-20 (Type I specimen) [32]. Testing was performed at both −40 °C and −60 °C. For each sample, five specimens were subjected to testing, and the results were reported as averages.

The determination of heat deflection temperature (HDT) adhered to the guidelines set forth by ASTM D648 standards [33], utilizing the CEAST HV6 HDT apparatus from Instron, USA. The HDT measurement entailed subjecting specimens measuring 127 × 12.7 × 3.2 mm to a load of 0.45 MPa and monitoring their deflection until it reached 0.25 mm. This deflection occurred as the specimens were gradually heated in a silicon oil bath, with the temperature increasing at a rate of 2 °C/min. For each sample, three specimens underwent testing, and the results were subsequently averaged and reported.

### 2.4. UV Weathering Test

To assess the impact of UV radiation on polymers in a swift and consistent manner, an accelerated UV test was employed as an expedited alternative. All test specimens underwent exposure to artificial weathering conditions for a duration of up to 200 h, utilizing weather-ometer equipment manufactured by ATLAS (Model Ci3000+). The testing protocol adhered to ASTM G155-21 standards [34]. This equipment incorporates a xenon lamp emitting at 4500 m, delivering a total of 100 cycles of irradiance at 0.35 W/m^2^ nm, specifically at a wavelength of 340 nm. The irradiance cycle involved 102 min of light exposure at a black panel temperature of 63 °C, followed by 10 min of light exposure accompanied by water spray, accumulating to a total of 100 cycles over 200 h. Subsequent to the UV irradiance cycle, the Izod impact test with notched specimens was conducted to assess the material’s performance. This evaluation was conducted in accordance with ASTM D256 [35], employing a hammer energy of 24.84 J at a temperature of 23 ± 2 °C and a relative humidity of 50 ± 5%. Five specimens were tested for each sample, and the results were averaged and reported.

### 2.5. Antimicrobial Activity Assessment

To evaluate the antimicrobial efficacy of the test specimens, a modified procedure was employed, deviating from both ASTM (ASTM E2149-20) [36] and JIS (JIS Z 2801:2000) standards [37]. Cultures of *Staphylococcus aureus (S. aureus*, ATCC 6538) and *Escherichia coli (E. coli*, ATCC 8739) were prepared from stock cultures and propagated in tryptic soy broth (TSB) to reach a final concentration of 2.5 × 10^5^ CFU/mL. Triplicate samples, each measuring 50 × 50 mm, were placed onto Petri dishes. Subsequently, each test specimen was inoculated with 400 μL of the bacterial suspension (10^5^ CFU/mL; OD_600_ = 0.4) and covered with a 40 × 40 mm piece of uncoated sterile PP film. Following a 24 h incubation period at 37 °C, bacteria from the film surface were collected using 10 mL of sterile saline. The collected suspension was subjected to serial dilution and plated on tryptic soy agar. To quantify the antimicrobial effect, the test specimens were also inoculated with 400 μL of the microorganism suspension and maintained at 35 °C for 24 h. After this incubation period, dilution and plating procedures were performed to determine the CFU/mL. The antimicrobial activity was expressed as a log reduction value, calculated using the following formula:Antimicrobial activity (log) = log(A) − log(B)(1)
where A represents the number of viable microorganisms in the control sample, and B corresponds to the number of viable microorganisms in the treatment sample.

### 2.6. Statistical Analysis 

Statistical analyses were executed using SPSS software (version 25, IBM, Chicago, IL, USA), employing ANOVA followed by Duncan’s multiple range tests to discern statistical distinctions among the mean values. Significance was determined at *p* < 0.05. The data are presented as mean ± standard deviation for each experimental dataset.

## 3. Results

### 3.1. Characterization

This study employed EDS (energy dispersive spectroscopy) and SEM (scanning electron microscopy) analyses to substantiate the incorporation of the antimicrobial agent DE/ZnO composite and the UV stabilizer in the HDPE injection mold. Figure 1 illustrates the distinct surface characteristics, revealing a notable contrast between raw HDPE and the injection mold containing the DE/ZnO composite. Notably, the latter exhibits a rougher surface texture. Furthermore, microscopic features measuring in micrometers are evident on the surface of the samples containing the UV stabilizer, unequivocally affirming the presence of a substantial quantity of T770 at the specimen surface. Elemental mapping was also employed to provide additional evidence of the DE/ZnO composite and T770 within the HDPE injection mold. Figure 2 displays the elemental mapping results of the test specimens. Notably, the presence of elements Zn and Si confirms the existence of the DE/ZnO composite, which serves as the antimicrobial material. Additionally, a notable increase in the characteristic element N is observed in correlation with the added amount of T770 in the test specimens. This elevation in N content primarily stems from the amine group within the chemical structure of T770, further corroborating its integration into the HDPE matrix [38,39,40].

Table 1 presents the outcomes of the Izod impact test and stability assessment under sub-zero temperatures. The results highlight that the incorporation of the DE/ZnO composite led to a notable enhancement in impact strength, while the stability in freezing temperatures remained virtually unaffected within the range of −40 °C to −60 °C when compared to pristine HDPE specimens. Interestingly, the introduction of a specific quantity of UV stabilizer did not exert any discernible influence on the impact strength and stability characteristics under sub-zero conditions. This phenomenon can be attributed to the inherent ductility of HDPE, which allows the DE/ZnO composite to distribute stress uniformly, thereby mitigating the likelihood of further fatigue-induced crack propagation [26,41].

Additionally, Table 1 offers a comprehensive overview of the mechanical properties of the tested specimens, including parameters such as tensile strength, elongation at break, and elastic modulus. The retention of HDPE’s mechanical properties, especially the elongation at break, is ascribed to the enhanced intermolecular and intramolecular chain interactions within the HDPE matrix when combined with the DE/ZnO composite. Furthermore, it can be inferred that the addition of T700 to the test specimens in various proportions yields similar mechanical properties as observed in the HDPE/DE/ZnO composite samples [42,43,44].

Heat deflection temperature (HDT) serves as a critical indicator of a composite material’s short-term resistance to elevated temperatures. This study adhered to ASTM D648 standards, subjecting test specimens to a flexural load of 0.455 MPa in the edgewise position. The HDT values established for all test specimens are summarized in Table 1. Notably, the HDPE/DZ blends exhibited a noteworthy increase in HDT when compared to pristine HDPE. This enhancement can be attributed to the high crystallization temperature of ZnO. Specifically, the inclusion of 4% wt (*w*/*w*) DE/ZnO in HDPE elevated the HDT from 75 °C (observed in neat HDPE) to 82 °C. It is noteworthy that the addition of T700 in the test specimens did not significantly impact the HDT when compared to HDPE/DZ samples. This heightened HDT is a direct consequence of the composite’s ability to resist deformation under load. Furthermore, the blending of HDPE with the DE/ZnO composite fosters intermolecular interactions between the molecular chains of the blended polymer [45,46,47].

The HDT results align with the findings from the previous TGA analysis (Figure 3), corroborating the relationship between the minimum volume fraction of particles incorporated into the composite and the comparable thermal properties of both the matrix and composites. This suggests that processing conditions akin to those employed for HDPE in an industrial-scale production scenario are feasible [48,49].

### 3.2. UV Weathering Test

Figure 4 and Figure 5 illustrate the impact resistance of the samples prior to and after UV irradiation at various time intervals. The Izod impact strength of the HDPE/DZ composite exhibits a modest drop, measuring at 3.4 kJ/m^2^ when comparing the test specimen of pristine HDDE at 5.8 kJ/m^2^. The inclusion of T700 has the potential to enhance the impact strength of HDPE/DZ test specimens. Based on the findings depicted in Figure 5, it can be observed that all the test specimens exhibit characteristics consistent with the hinged break type, indicating an incomplete fracture. The observed reduction in impact resistance in HDPE/DZ test specimens may be attributed to the influence of UV radiation, which is susceptible to the effects of moisture and photooxidation during rapid weathering. These factors might lead to the fragmentation of the polymer chain inside the material. Furthermore, the presence of a UV stabilizer effectively hinders the production of hyperoxide groups, hence preserving the impact strength of the sample containing T700 regardless of the quantity present [50,51,52].

### 3.3. Investigation of Antibacterial Properties in Injection-Molded Test Specimens

The study delves into the antibacterial characteristics of injection-molded test specimens. Neat HDPE samples, intriguingly, did not exhibit inhibition against either *S. aureus* (Figure 6) or *E. coli* (Figure 7). However, the inclusion of HDPE specimens significantly bolstered antibacterial activity, manifesting as a notable increase of 2.7 and 2.8 log units in response to *S. aureus* and *E. coli,* respectively (Table 2). In the case of test specimens containing T700, there was a slight decrease in antimicrobial activity across all concentrations, with approximately consistent values of log 2.2 against *S. aureus* and log 2.6 against *E. coli.* This signifies a marginal reduction in antimicrobial performance following the addition of T700. Nevertheless, the performance remains at a high level, rendering it suitable for utilization in the seafood product industry.

The study’s findings shed light on the mechanism behind the antibacterial action of the ZnO composite, corroborating prior research. The antibacterial effect of metal particles involves multiple processes, encompassing the generation of reactive oxygen species, the release of cationic ions, and cell wall disruption [52,53,54,55]. It is well-documented that ZnO and Zn^2+^ ions can penetrate bacterial cell walls, interacting with cytoplasmic components to effectively combat bacteria. This interaction elucidates the dynamic interplay between the ZnO composite and bacteria at the interface. Notably, bacterial cell walls carry a negative surface charge. The antibacterial activity of the DE/ZnO composite can establish or alter electrostatic interactions between ZnO particles and bacterial cell membranes [18,26,56,57,58].

This study also delves into the antimicrobial properties of seafood product packaging crafted from HDPE/DZ/T-500. The choice of HDPE/DZ/T-500 for manufacturing stems from its notably high antimicrobial activity and minimal UV stabilizer content, rendering it both cost-effective and highly efficient. To assess applicability, HDPE/DZ/T-500 containers were deployed in a bustling fish market in Busan, Republic of Korea. Samples were retrieved after various periods of use, spanning 3 months, 6 months, and 1 year. The outcomes, presented in Table 3, reveal a significant decline in antimicrobial activity after a year of utilization, with residual antimicrobial effectiveness against *S. aureus* and *E. coli* at 90.68% and 89.81%, respectively.

## 4. Conclusions

This study extended its investigation to assess the antimicrobial efficacy of seafood product containers manufactured from HDPE/DZ/T-500. The choice of HDPE/A/TNV-500 for container production was deliberate, owing to its elevated antimicrobial activity and minimal UV-stabilizer content, representing a cost-effective and high-performance option. These HDPE/DZ/T-500 containers were deployed in a practical setting at a bustling fish market in Busan, Republic of Korea. Samples were methodically collected following periods of active use, spanning 1 month, 3 months, 6 months, and 1 year. The results reveal a noteworthy decline in antimicrobial activity after a year of utilization, with residual antimicrobial effectiveness observed against *S. aureus* at 90.68% and *E. coli* at 89.1%. This investigation underscores the impact of prolonged container use, unveiling degradation in both physical and mechanical properties. The degradation witnessed has a significant impact on the antimicrobial efficacy of the containers, leading to the observed reduction in effectiveness. To sum up, these discoveries highlight the evolving nature of container performance with the passage of time, emphasizing the importance of regular evaluation and upkeep in real-world usage. This research offers valuable perspectives into the intricate relationship between the material characteristics of containers and their antimicrobial performance within practical seafood product packaging contexts.

## Figures and Tables

**Figure 1 polymers-15-04577-f001:**
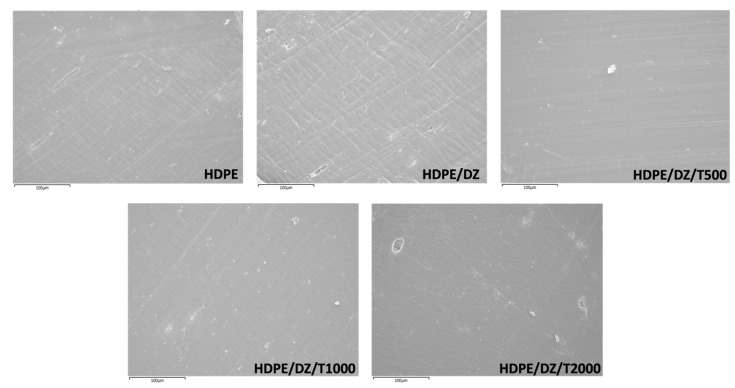
FE-SEM images of test specimens.

**Figure 2 polymers-15-04577-f002:**
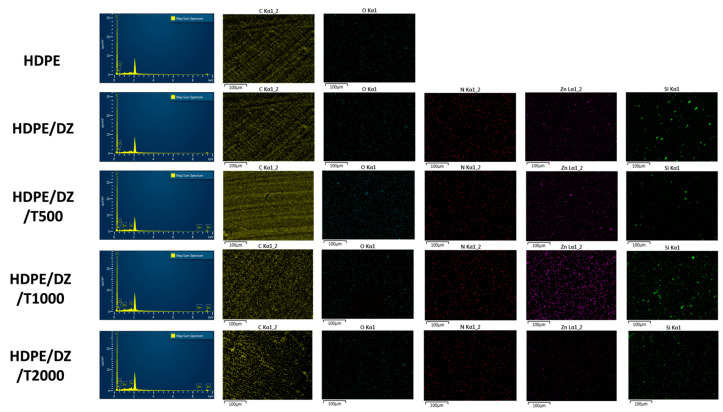
EDS mapping images of test specimens.

**Figure 3 polymers-15-04577-f003:**
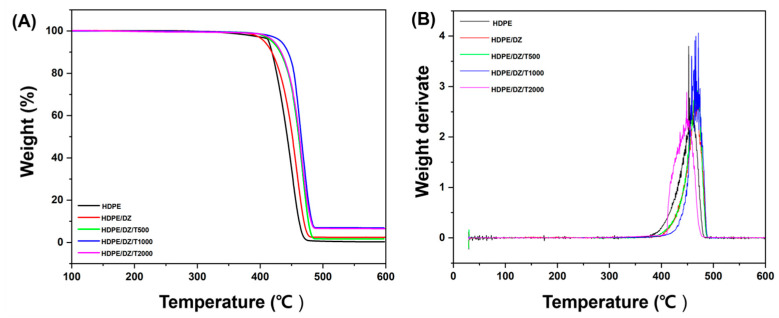
Thermogravimetric analysis (**A**) TGA and (**B**) DTG of test specimens.

**Figure 4 polymers-15-04577-f004:**
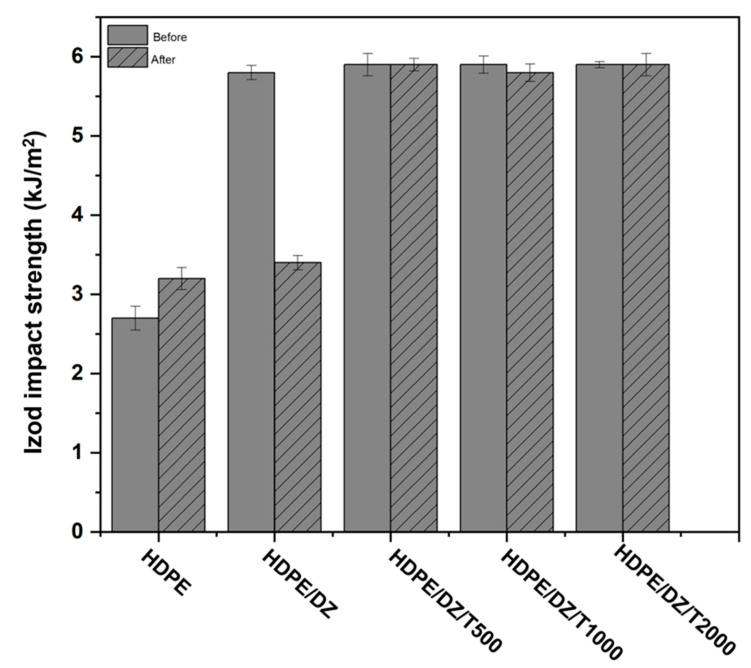
Izod impact strength values of test specimens before and after weathering test.

**Figure 5 polymers-15-04577-f005:**
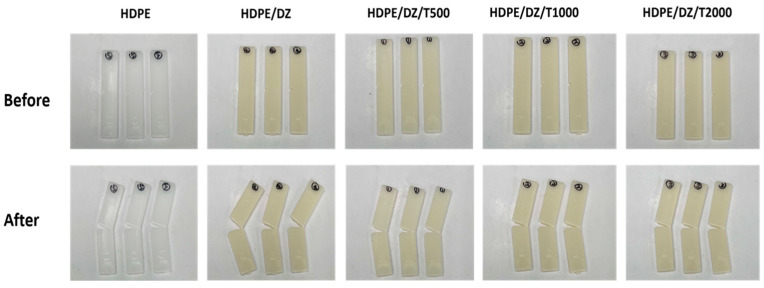
Results of the accelerated weathering test on test specimens.

**Figure 6 polymers-15-04577-f006:**
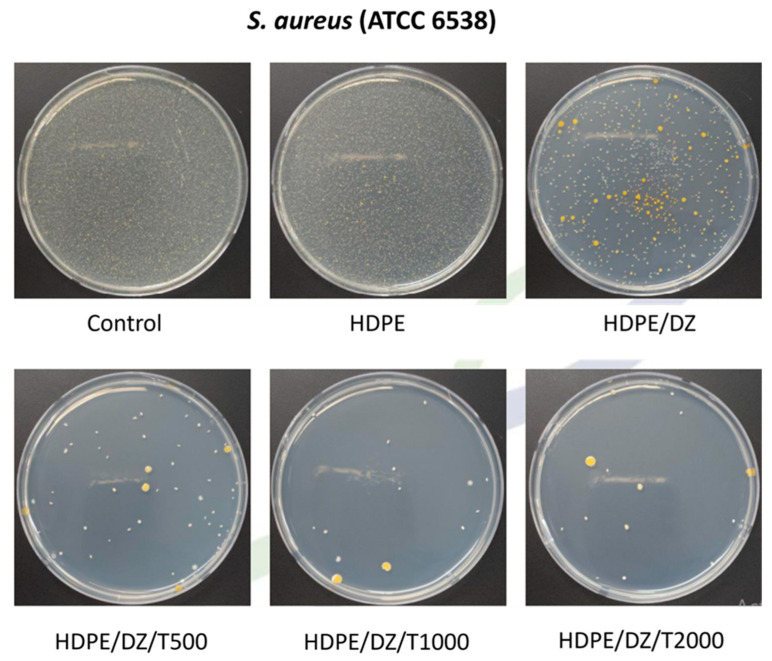
Antimicrobial activity of test specimens against *S. aureus*.

**Figure 7 polymers-15-04577-f007:**
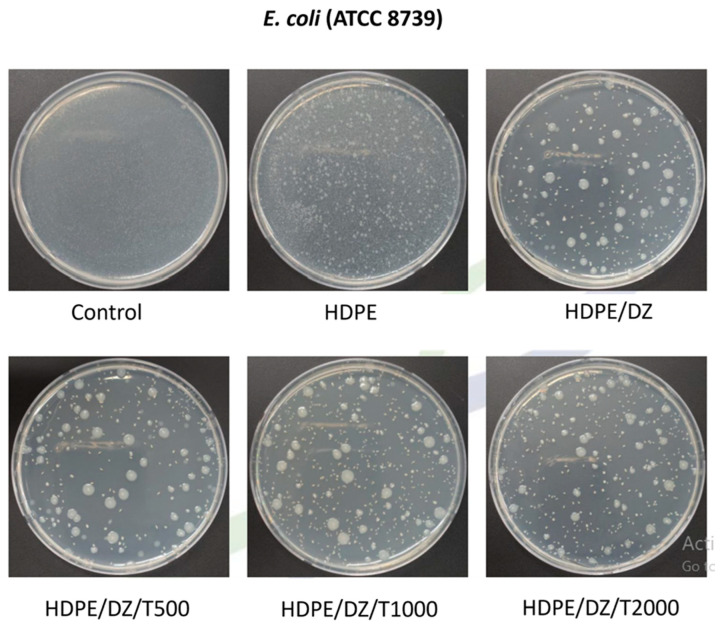
Antimicrobial activity of test specimens against *E. coli*.

**Table 1 polymers-15-04577-t001:** Material properties and sub-zero temperature performance of HDPE-based composites.

Sample	Izod Impact (J/m)	Stability at Sub-Zero Temperature		Tensile Strength (MPa)	Elongation (%)	Elastic Modulus (MPa)	HDT (°C)
		−40 °C	−60 °C				
HDPE	53.45 ± 1.09 ^b^	×	×	26.23 ± 0.21 ^a^	53.14 ± 4.32 ^a^	0.51 ± 0.42 ^b^	75.40 ± 1.53 ^b^
HDPE/DZ	163.15 ± 3.28 ^a^	O	O	27.31 ± 1.52 ^a^	48.19 ± 1.75 ^b^	0.64 ± 0.87 ^a^	82.64 ± 1.34 ^a^
HDPE/DZ/T500	164.43 ± 5.12 ^a^	O	O	28.49 ± 1.81 ^a^	47.61 ± 1.98 ^b^	0.65 ± 0.11 ^a^	81.45 ± 0.89 ^a^
HDPE/DZ/T1000	164.12 ± 1.81 ^a^	O	O	28.15 ± 1.09 ^a^	47.84 ± 1.02 ^b^	0.65 ± 0.21 ^a^	82.35 ± 1.45 ^a^
HDPE/DZ/T2000	164.71 ± 2.53 ^a^	O	O	28.63 ± 1.17 ^a^	47.13 ± 0.98 ^b^	0.64 ± 0.93	82.71 ± 0.83 ^a^

Note: values in the same column, denoted by different letters, exhibit significant differences (*p* < 0.05) as determined by Duncan’s multiple range tests. (×) signifies that the test specimen undergoes alterations during testing, while (O) indicates its stability is maintained.

**Table 2 polymers-15-04577-t002:** Antimicrobial activity against *S. aureus* and *E. coli*.

Sample	HDPE	HDPE/DZ	HDPE/DE/T500	HDPE/DE/T1000	HDPE/DE/T2000
Antimicrobial Activity (%)					
*S. aureus*	0.00 (log 0.0)	99.80 (log 2.7)	99.40 (log 2.2)	99.38 (log 2.2)	99.37 (log 2.2)
*E. coli*	0.00 (log 0.0)	99.85 (log 2.8)	99.78 (log 2.7)	99.78 (log 2.7)	99.75 (log 2.6)

**Table 3 polymers-15-04577-t003:** Antimicrobial activity of HDPE/DZ/T500 composite over time against *S. aureus* and *E. coli*.

Sample (HDPE/DZ/T500)	Day 0	Day 30	Day 90	Day 180	Day 365
Antimicrobial Activity (%)					
*S. aureus*	99.40 (log 2.2)	98.76 (log 1.90)	98.59 (log 1.85)	98.18 (log 1.74)	90.68 (log 1.03)
*E. coli*	99.78 (log 2.7)	98.92 (log 1.97)	98.15 (log 1.73)	98.03 (log 1.71)	89.81 (log 0.99)

## Data Availability

Data are contained within the article.

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
