# Peer review of "Innovative HDPE Composites Enriched with UV Stabilizer and Diatomaceous Earth/Zinc Oxide for Enhanced Seafood Packaging and Antimicrobial Properties"

_polymers, 2023, doi:10.3390/polym15234577_

Round 1
Reviewer 1 Report
Comments and Suggestions for Authors
Peer reviewers have positively evaluated the study Innovative HDPE Composites Enriched with UV Stabilizer and Diatomaceous Earth/Zinc Oxide for Enhanced Seafood Packaging and Antimicrobial Properties highlighting the proper characterization and logical results with appropriate comparisons to previous works. However, some minor revisions are needed before publication.
1.Materials and methods UV stabilizers: How these ratios were selected?
2. No reference is mentioned for antimicrobial activities kindly include the reference given below (Antimicrobial efficacy of clove essential oil infused into chemically modified LLDPE film for chicken meat packaging)
3. standard deviations are not mentioned in Table 2 and 3.
4. statistical analysis of the results was not mentioned in the manuscript.
5. Figure 1 images are not clear use higher dpi for it.
6. conclusion and abstracts need some modifications.
Author Response
The response report has been attached as a file

Reviewer 2 Report
Comments and Suggestions for Authors
The manuscript titled ‘Innovative HDPE Composites Enriched with UV Stabilizer and Diatomaceous Earth/Zinc Oxide for Enhanced Seafood Packaging and Antimicrobial Properties' presents some interesting results relevant to active food packaging. The manuscript is well-written and the study is well-designed. Below are some suggestions to make the manuscript better.
1. SEM images do not indicate substantial evidence of additive substances in the film. Have you tried with a higher magnification? The images provided are too small to be viewed clearly.
2. Lines 185-190 present a very confusing status on the film's mechanical properties. First, the authors claim the amplified intermolecular and intramolecular chain interactions within the HDPE matrix when combined with the DE/ZnO composite preserve the mechanical properties and then claim that the incorporation of both DE/ZnO and T700 has no substantial impact on the mechanical properties of HDPE. There is a decrease in the elongation and an increase in the Elastic modulus. How do you infer these results? Please elaborate on this.
3. Please indicate the acceptable levels of microbial contamination by the standard organization and compare them with your obtained results
4. Lines 249-252, please substantiate the claims by performing the ROS assays.
5. Line 260-261, What is referred to as ¨Containers¨ here? And how is the film appropriate for that?
6. Did you compare the antimicrobial activity of your film with any control sample for the long-term storage study? How can these results be compared unless there is no control?
Author Response

(The authors gave the same response as above.)
